# Inflammatory Response in Caco-2 Cells Stimulated with *Anisakis* Messengers of Pathogenicity

**DOI:** 10.3390/pathogens11101214

**Published:** 2022-10-20

**Authors:** Ilaria Bellini, Daniela Scribano, Meysam Sarshar, Cecilia Ambrosi, Antonella Pizzarelli, Anna Teresa Palamara, Stefano D’Amelio, Serena Cavallero

**Affiliations:** 1Department of Public Health and Infectious Diseases, Sapienza University of Rome, 00185 Rome, Italy; 2Research Laboratories, Bambino Gesù Children’s Hospital, IRCCS, 00146 Rome, Italy; 3Human Sciences and Promotion of the Quality of Life, San Raffaele Open University, IRCCS, 00166 Rome, Italy; 4Laboratory Affiliated to Institute Pasteur Italia—Cenci Bolognetti Foundation, Department of Public Health and Infectious Diseases, Sapienza University of Rome, 00185 Rome, Italy; 5Department of Infectious Diseases, National Institute of Health, 00185 Rome, Italy

**Keywords:** *Anisakis*, Caco-2, crude extract, extracellular vesicles, inflammation

## Abstract

**Background**: *Anisakis* spp. third-stage larvae (L3) are the causative agents of human zoonosis called anisakiasis. The accidental ingestion of L3 can cause acute and chronic inflammation at the gastric, intestinal, or ectopic levels. Despite its relevance in public health, studies on pathogenetic mechanisms and parasite-human interplay are scarce. The aim of this study was to investigate the human inflammatory response to different *Anisakis* vehicles of pathogenicity. **Methods**: Human colorectal adenocarcinoma (Caco-2) cells were exposed to *Anisakis* L3 (the initial contact with the host), extracellular vesicles (EVs, *Anisakis*–host communication), and crude extract (CE, the larval dying). The protein quantity and gene expression of two pro-inflammatory cytokines (IL-6 and IL-8) were investigated using an ELISA test (6 h and 24 h) and a qReal-Time PCR (1 h, 6 h, and 24 h), respectively. **Results**: The L3 and EVs induced a downregulation in both the *Il-6* and *Il-8* gene expression and protein quantity. On the contrary, the CE stimulated IL-6 gene expression and its protein release, not affecting IL-8. **Conclusions**: The Caco-2 cells seemed to not react to the exposure to the L3 and EVs, suggesting a parasite’s immunomodulating action to remain alive in an inhospitable niche. Conversely, the dying larva (CE) could induce strong activation of the immune strategy of the host that, in vivo, would lead to parasite expulsion, eosinophilia, and/or granuloma formation.

## 1. Introduction

Nematodes of the genus *Anisakis* are cosmopolitan parasites that depend on numerous aquatic hosts to successfully complete their life cycle [1]. Their occurrence as third-stage infective larvae (L3) in marine products such as fish and cephalopods intended for human consumption is of economic and medical concern [2], enough to define *Anisakis* as an emerging pathogen. Despite that *Anisakis* L3 cannot reach the adult stage in humans, the ingestion of a single L3 can accidentally determine a fish-borne zoonosis called anisakiasis. Human anisakiasis is a disease characterized by a range of unspecific symptoms leading to acute or chronic forms [3], often associated with clinical signs similar to those observed with bacterial or viral gastroenteritis (diarrhea, nausea, and abdominal pain). It shows mild to severe gastric and intestinal signs and/or allergic reactions (urticaria, rhinitis, and anaphylaxis) [4]. These are a result of the combined action of inflammatory/immunological responses, mechanical damage caused by the larval migratory behavior through the gastro-intestinal mucosa [5], and the release of secreted serine and metalloproteases that degrade collagen and glycoproteins [6]. If not removed or expelled, the larva dies a few weeks after ingestion [7]. Therefore, its dying involves the release of somatic antigens [8,9] able to cause chronic lesions such as granulomas, persistent inflammation, and eosinophilic infiltration [7].

The incidence of anisakiasis is strictly related to the traditional food habit of eating raw fish, and Japan alone accounts for around 90% of the total reported cases [10]. The remaining cases occur in countries such as Korea, China, Peru, the Netherlands, Germany, France, Spain, Croatia, and Italy [11,12,13,14]. In Italy, around 400 symptomatic cases have been reported in hospital discharge records in a decade [15]. Given several not-exclusive factors such as the large occurrence of L3 in numerous fishes, the emerging trend of exotic dishes, the increasing fish consumption worldwide, the difficulty in diagnosis, and the asymptomatic cases, anisakiasis remains largely underestimated [16].

In addition, the association between *Anisakis* and tumorigenic potential is also not completely understood. Helminths such as flukes (among several, including *Schistosoma haematobium, Opisthorchis viverrini,* and *Clonorchis sinensis*) are capable of significant tumor-promoting activity [17,18,19]. Reports of tumor co-localization with *Anisakis* L3 and L3 mimicking metastatic lesions are increasing [20,21,22] and, interestingly, cases have been reported mainly in countries with a higher prevalence of anisakiasis [23,24]. Despite that it is still unclear if anisakiasis and carcinoma are causatively related or accidental incidences [25], neoplasia and embedded larvae share a common site in all of the reported cases [26]. Nonetheless, exposure to *Anisakis* is suggested as a risk factor for gastric or colon adenocarcinoma [27] but to date, data about the pathogenetic mechanisms are very scarce, and in this scenario, investigations about human-*Anisakis* interactions have acquired even greater importance. 

Among the parasitic aspects involved in human-*Anisakis* interplay, three main factors can be identified: the live larvae itself, the entire antigenic content of the whole parasite (namely the crude extract), and the excreted/secreted molecules (E/S), including extracellular vesicles (EVs), in particular, exosomes. EVs have been recently detected in helminths, including nematodes [28,29,30], revealing a new paradigm in host–parasite relationships. EVs may represent a new potential vehicle of pathogenicity, whose content is characterized by transporting key elements such as miRNAs that are involved in a complex regulatory network modulating gene expression [31]. This packaged content suggests a strong immunomodulatory potential, able to suppress the host’s Type 2 innate immune response in vivo [32] to ameliorate host inflammatory pathways [33] and to modulate the transcription of related target genes, affecting the human immune system [30]. To date, most of the data about anisakiasis are based on in vitro models that do not fully resemble the microenvironment in which the parasite acts. However, available studies using human fibroblasts [34], human dendritic cells [35], and human epithelial colonic cancer cells (Caco-2) [36] described a modulatory activity exerted by *Anisakis* parasitic products, consisting of the upregulation of oxidative stress, inhibition of apoptosis-related biomarkers, and inflammatory response.

Given this scarce availability of information on the disease and its potential consequences, the present study attempts to extend the knowledge in this field using Caco-2, a suitable in vitro model, to recreate the parasite–host interactions, given its ability to differentiate into a monolayer characterized by many typical properties of the large intestines, despite that they originally derived from colon carcinoma [37]. According to that, this model has been widely used for viral, bacterial [38,39], and parasitic infections [40,41,42].

Here, Caco-2 cells have been exposed to three *Anisakis*-derived biological samples: (i) the active live infective larva (=L3), as a model for the initial contact with the intestinal epithelium; (ii) the crude extract (=CE), representing the whole body of senescent larvae, mimicking the moment in which the parasite, failing to complete its life-cycle, and eventually dying, and (iii) the extracellular vesicles enriched fraction (=EVs), as the potential vehicle of pathogenicity. In particular, the gene expression and the protein product abundance of two crucial pro-inflammatory cytokines (IL-8 and IL-6) have been assessed by a quantitative real-time PCR (qRT-PCR) and a sandwiched enzyme-linked immunosorbent assay (ELISA), respectively. IL-8 is a potent chemokine secreted by different cell types including blood monocytes, fibroblasts, and epithelial and endothelial cells, and it is involved in the recruitment of neutrophils and granulocytes to the site of damage [43]. IL-6, as a pleiotropic cytokine with a critical role in inflammation and hematopoiesis, acts as a link between the innate and acquired immune response, and it is often involved in autoimmune disease and tumorigenesis [44].

## 2. Results

### 2.1. Parasite Samples and Identification

A total of 485 L3 were collected and identified following a restriction analysis of the nuclear ribosomal, ITS with *Hinf*I. The enzymatic cuts produced different banding patterns: three fragments of ∼370, 300, and 250 bp in *A. pegreffii*; two fragments of ∼620 and 250 bp, plus one additional band at 80 bp in *Anisakis simplex* sensu stricto, and all of the four fragments in the hybrid genotype of the two mentioned species. Once identified, the L3 were used for further experiments: 33 (46% *A. pegreffii*, 50% *A. simplex* s.s., and 4% putative hybrids) for the challenge with the live parasites, 72 (70% *A. pegreffii*, 17% *A. simplex* s.s, and 13% putative hybrids) for the crude extracts, and 380 (61% *A. pegreffii*, 23% *A. simplex* s.s, and 16% putative hybrids) for the EVs-enriched fraction isolation. 

### 2.2. EVs-Enriched Fraction Characterization 

Nanoparticle tracking analysis (NTA), using Nanosight technology, estimated the size distribution and concentration of the particles in the exosomal-enriched fraction (Appendix A). The mean size of the particles in the *Anisakis* EV samples was 141.7 nm (mode, 107.4). Three main numbers of peaks were obtained (main peak, 115 nm), and a concentration of 1.32 × 10^10^ particles/mL was reported (Figure 1).

A Western blot with an anti-Alix antibody, performed on the *Anisakis* EV-enriched fraction, the *Anisakis* CE, and on the Caco-2 cells as the controls, showed a band at 110 kDa in the positive controls (Caco-2), while in the CE and EVs, the bands were specific but at a higher molecular weight of around 130 kDa (Appendix A), probably due to post-transcriptional modifications of the protein.

### 2.3. Intestinal Epithelial Cells’ Response to Live L3 Actions

An additional time point at 6 h was included in the incubation with the L3, in order to monitor its potential mechanical activity on the cells’ monolayer integrity. This time point has been analyzed by ELISA, too.

No morphological changes, such as cytoplasmic vacuolization, shrinkage, plasma membrane blebbing, and chromatin condensation, were observed in the Caco-2 cells after 1 h, 6 h, and 24 h post-contact with the *Anisakis* products or in the unexposed Caco-2 cells. The challenge with the live L3 showed a progressive decrease of IL-6 in the Caco-2 cells incubated at 6 h (not significant) and 24 h (F(2,37) = 8.143, *p* = 0.0001) if compared with the non-treated cells. In order to evaluate the gene expression and cellular response over time, qRT-PCR analyses were performed at three different time points (GAPDH Eff: 99.97%, *Il-8* Eff: 98.71%, and *Il-6* Eff: 90.59%). However, the results obtained for the *Il6* gene expression showed no significant changes. On the other hand, levels of the neutrophil chemotactic factor IL-8 (Figure 2) were consistently reduced in the Caco-2 cells incubated with the live L3 for 6 h (F(2,37) = 17.85, *p* < 0.0001), if compared to the controls, followed by a significant increase at 24 h (6 h vs. 24 h: *p* < 0.01). However, the IL-8 levels at 24 h did not significantly differ from the controls. Similarly, the gene expression of *Il8* showed a slight decrease at 1 h compared to the non-treated cells (not significant), followed by an increase at 6 h (F(2,33) = 3.060, *p* = 0.0422, 1 h vs. 6 h: *p* < 0.05) that tended to reach the level of expression as in the controls (controls vs. 6 h = ns, and controls vs. 24 h = ns) (Figure 2).

### 2.4. Intestinal Epithelial Response to Anisakis EVs 

The challenges of the Caco-2 cells with the *Anisakis* EVs revealed a decreasing trend for the two cytokines of interest. In particular, IL-6 was not detected in the ELISA assay compared to the controls (t(29) = 3.42, *p* = 0.0065). The relative quantification of the *Il*6 gene expression showed a statistically relevant (F(3,18) = 9.205, *p* = 0.0007) upregulation at 1 h (*p* < 0.001), followed by a significant decrease in gene expression at 6 h and 24 h (1 h vs. 6 h: *p* < 0.05; 1 h vs. 24 h: *p* < 0.01, respectively), reaching the expression level reported in the controls (Figure 3). A similar pattern was also observed for the IL-8 assays. IL-8 in the Caco-2 cells treated with the EVs after 24 h appeared decreased (t(31) = 2.92, *p* = 0.0127) compared to the controls, and the *Il*8 gene expression analysis showed a statistically relevant (F(3,20) = 5.762, *p* = 0.0052) upregulation at 1 h (*p* < 0.05) that decreased at the other checkpoints (1 h vs. 6 h: *p* < 0.05; 1 h vs. 24 h: *p* < 0.05), reaching the control levels of the gene expression (Figure 3).

### 2.5. Intestinal Epithelial Cells’ Response to Anisakis Crude Extract

The number of proteins observed in the CE samples used for the Caco-2 cells challenge was about 900 ng/µL. The Caco-2 cells exposed to the *Anisakis* CE for 24 h showed a strong, relevant increase in IL-6 secretion (t(22) = 3.64, *p* = 0.0054) compared to the control group (Figure 4). This evidence was also supported by the qRT-PCR analyses showing an increasing trend in the *Il6* gene expression at 1 h (not significant) and 6 h (F(3,20) = 8.143, *p* = 0.0010). Then, the *Il*6 expression decreased significantly (6 h vs. 24 h: *p* < 0.05), reaching the expression level reported in the controls (Figure 4). Interestingly, the CE slightly affected the IL-8 secretion, showing a mild but not significant decrease in the ELISA test compared to the controls (t(24) = 1.49, *p* = 0.1612) (Figure 4). According to that, a slight decrease in the *Il*8 gene expression was observed at 6 h, with respect to the other checkpoints analyzed (1 h and 24 h), however, without statistical support (F(3,20) = 2.805, *p* = 0.0661) (Figure 4).

## 3. Discussion

Helminths, multicellular worm parasites including nematodes, cestodes, and trematodes, are able to establish long-lasting chronic infections, indicating the successful manipulation of the host immune system, as they need to survive in inhospitable environments during infections. A co-evolutive adaptation is not likely to be applicable to anisakiasis, as humans are accidental hosts; however, despite studies aimed to explore *Anisakis* ability in modulating humans’ inflammatory and immune response increasing [35,45,46], the fine molecular mechanisms related to its pathogenesis and clinical outcomes are still mostly obscure.

Many parasitic nematodes cannot be easily cultured, and existing in vitro and in vivo models do not sufficiently recount the human background and disease. To date, only two studies used Caco-2 cells to investigate human anisakiasis, without a focus on the cytokines’ release [36,37,38,39,40,41,42]. Hence, the present investigation was performed using Caco-2 cells as a suitable in vitro model able to recreate a microenvironment for *Anisakis* infection. Accordingly, we tested three challenges mimicking two pivotal phases of anisakiasis: (i). active penetration of the tissue (L3) and communication with the intestinal epitheliums (EVs) and (ii). the larval dying (CE). The present study is the first attempt to explore the cellular response after exposure to the EVs derived from the infective third-stage larvae of the zoonotic nematode *Anisakis* spp. (*A. pegreffii*, *A. simplex* s.s., and their hybrid form).

The evidence, here collected about inflammatory pathways after challenges with *Anisakis*-derived products, revealed interesting and variable outcomes of the cytokines’ production and gene expression. A few hours after ingestion, the L3 release of proteolytic enzymes and chemotactic factors to adhere to and penetrate the mucosa and submucosa [47,48] induced hemorrhagic and erosive lesions [7] associated with a production of excretory/secretory factors able to interact with the human first line of defense [49]. The first cells to encounter the invading *Anisakis* L3 were the epithelial cells, a physical barrier also equipped with microbial-detection mechanisms, signaling the circuits and inflammatory mediators [50]. Surprisingly, the present data showed that the Caco-2 cells were not significantly triggered by the presence of the live L3. Hence, the decrease in the IL-6 and IL-8 cytokines’ quantity that was observed could, therefore, reflect a first strategy by which the parasite modulates the epithelial barrier response triggered by the initial contact with the host, quickly returning the system to homeostasis and keeping the host healthy for successful long-lived infections. Such a concept was also described in the phylogenetic-related ascaridoid *Ascaris suum* [51], in which the transcriptional analysis of L3 incubated with porcine intestinal epithelial cells demonstrated a low magnitude of inflammatory response characterized mainly by IL-8 and NF-kB suppression. Our results, based on ELISA and qRT-PCR analyses, showed live L3 inducing an early and transient downregulation of the host cytokines’ gene expression and protein release. Moreover, the effect of the IL-6 amount was longer than that of IL-8. Despite the significant reduction at 6 h, IL-8 turned to the basal level at 24 h. This is in agreement with a previous study in which IL-8 mRNA alterations were not detected in patient sera exposed to *Anisakis simplex* [52]. To our knowledge, only Napolitano et al., 2018, have investigated IL-6 secretion in the *Anisakis* L3-human interaction. Immature and mature dendritic cells (DCs) incubated with seven live specimens of L3 for six days showed an increase in IL-6 secretion and a less reactive phenotype of DC not sufficient to drive a Th2/Th17 immune response [35]. In the present study, consistent outcomes about a suppressive action of live L3 on the first line of human defense were obtained, in this case, based on the decrease of the IL-6 amount.

Besides the live L3, their derived EVs are an additional important factor to consider in the early pathogenicity exerted by *Anisakis*. EVs are a newly discovered vehicle of early communication between the host and pathogens, since they determine strong immunomodulatory effects [53], as demonstrated for other parasitic nematodes [29,30,32,54,55]. *Anisakis* live L3 are able to release EVs, as recently demonstrated [56], also, in the same experimental condition here performed [57]. The EVs-enriched fraction used in the present study is the result of the incubation of numerous larval specimens (a pool of 16 to 20 larvae), while usually a single L3 infects a human host. Consequently, a stronger activity of such EVs-enriched fraction in comparison with a single larva is expected. According to this hypothesis and to published data [32,54], we did not detect IL-6 and we observed a decrease in IL-8 in the ELISA at 24 h. However, the expression of the *Il*6 and *Il*8 genes showed a different trend. The exposure to EVs induced an early upregulation of both genes, followed by a progressive downregulation over time, even if not significant, compared to the basal level of gene expression in the controls. Based on these results, the attenuation of IL-6 and IL-8 may have occurred via a post-transcriptional mechanism, potentially due to proteases, as observed in other intestinal parasites [58] and usually released also by *Anisakis* [47], or to other potential regulatory factors such as miRNAs transported in EVs. Alternative explanations may be related to the time points selected that may not be representative of the precise time of the cytokines’ mRNA downregulation. A crucial factor to consider is the timing of the EVs’ uptake by the cells. *Heligmosomoides polygyrus* EVs are better internalized by epithelial cells (MODE-K) at 24 h (55%) compared to 1 h (10%) [54]; *Trichuris muris* EVs are internalized by murine colonic organoids within 3 h at 37 °C [30], and *Anisakis*-derived EVs are internalized by human macrophage-Like THP-1 cells, but a precise indication of the uptake timing is lacking [56]. In this scenario, the early upregulation of the *Il*6 and *Il*8 gene expressions, here observed, could be mainly due to the interaction between the host cellular receptors and the *Anisakis* extracellular EVs’ surface proteins, rather than a function associated with the EVs’ content. Consequently, the observed immunomodulatory effect exerted by live L3-releasing EVs on intestinal cells may be characterized by several not-exclusive mechanisms of action.

A different scenario is detectable in the CE challenge regarding IL-6 production. CE represents a mixture of L3 components (endotoxins, proteases, metalloproteases, and somatic proteins) that may be clinically compared to the moment of L3 death (around 14 days after ingestion) associated with the parasite’s expulsion or granuloma formation and chronic inflammation [7]. Available in vitro studies described a pro-inflammatory activity exerted by *Anisakis* parasitic products, including the upregulation of oxidative stress, the barrier’s integrity alteration, inhibition of apoptosis-related biomarkers, and inflammatory induction [34,35,36,42]. According to that, our data about Caco-2 cells exposed to CE showed a strong upregulation of IL-6 release and in the *Il6* gene expression. This pro-inflammatory enhancement could represent a trigger for the host immune response, able to activate and recruit immune cells (macrophages, DCs, eosinophils, and neutrophils) to the site of damage, determining the arrest or the disease progression (chronic inflammation and granulomas). Additionally, such data are in agreement with those obtained by previous in vivo and in vitro studies on *Anisakis* CE [34,35,59]. These studies highlighted a pathological condition related to chronic inflammation and, in turn, to a potential progressive increased risk of the host’s DNA damage and cancer, as demonstrated for other helminths [60]. To date, only two studies have investigated these neglected aspects in the framework of anisakiasis. The tumorigenic potential of *Anisakis* was explored using hamster ovary cells and Sprague–Dawley rats, revealing an increase in cell proliferation, a reduction of apoptosis, and changes in the expression of serum cancer-related miRNAs in rats [59]. Moreover, an increased level of P53 and ROS in the fibroblast cell line, HS-68, treated with *Anisakis* excretory/secretory products and CE was observed [34]. Interestingly, our results showed that CE did not particularly affect the *Il*8 gene expression and secretion by Caco-2 cells, suggesting IL-8 as a non-crucial factor involved in anisakiasis. Numerous parasite-derived factors have been reported to recruit, often selectively, neutrophils or eosinophils, and the somatic extracts of *A. simplex* did not elicit a measurable chemotactic response in the neutrophils, as previously suggested [61]. 

## 4. Materials and Methods

### 4.1. Parasites Sampling

*Anisakis* live third-stage infective larvae (L3) were collected from the visceral cavity of 21 European hakes *Merluccius merluccius* purchased at markets between 2018 and 2021 and captured from area FAO 37. L3 were washed three times in a 0.22 µm filtered phosphate buffer saline (PBS). After the experiments, all the L3 used for the study were stored at −20 °C for subsequent identification of species through a molecular diagnostic key based on the PCR-RFLP of the internal transcribed spacers of the nuclear ribosomal DNA, according to D’Amelio et al., 2000 [62]. 

### 4.2. Crude Extract 

A pool of L3 (*n* = three) was used for obtaining crude extract samples. CE was prepared as follows: PBS 10X was added to each sample and then homogenated with pestles. The mixture (0.13 mg/mL) was centrifuged at 13,500 rpm for 30 min. at 4 °C, according to Mattiucci et al., 2017 [63]. Lastly, the supernatant was used for the challenge with Caco-2 cells, whereas the pellets were stored in 70% ethanol for the subsequent identification of species. The protein concentration of samples was evaluated using a Qubit4 (Thermo Fisher Scientific, Waltham, MA, USA).

### 4.3. Isolation and EVs-Enriched Fraction Characterization 

Pools of L3 (*n* = 20) were incubated in RPMI with 1X Pen/Strep for 24 h at 37 °C with 5% CO_2_. After incubation, the culture media were collected to isolate the exosomal-enriched fraction of the EVs using an ExoQuick kit (System Biosciences, Palo Alto, CA, USA) in accordance with the manufacturer’s protocol. The samples obtained were diluted in a 0.22 µm filtered PBS and immediately used for incubation with Caco-2 cells, while the larvae were stored for the subsequent identification of species. 

The size distribution and concentration of particles in the recovered fractions after the exosome enrichment were measured using nanoparticle tracking analysis (NTA), using a Nanosight NS300 (Malvern Panalytical, Malvern, UK). Five measurements were performed with a 60 s duration for each measurement and the data were analyzed using NTA software version 3.4. 

To further characterize the presence of extracellular vesicles in our samples, Western blotting analysis of the isolated EVs against the cytosolic EVs’ marker ALG-2-interacting protein X (Alix) antibodies were undertaken. The protein concentration of the EV samples was evaluated using a Qubit4 (Thermo Fisher Scientific, Waltham, MA, USA). The CE sample (used as the larval control), Caco-2 cells (used as a positive control for the antibody), and the EV samples were lysed in a 5X (CE and EVs) and 1X (Caco-2 cells) sample buffer (50 mMTris–HCl pH 6.8, 2% SDS, 10% glycerol, 5% β-mercaptoethanol). About 8 µg of proteins were resolved on a 12% Tris–glycine SDS-PAGE and transferred onto nitrocellulose membranes (Hybond-P GE-Healthcare, Sigma-Aldrich St. Louis, MO, USA). Western blot analyses were performed using polyclonal, rabbit anti-Alix (1:500, TBS-T, 5% NFDM) from Thermo Fisher Scientific, USA (Catalog # PA5-52873). Anti-rabbit antibody conjugated to horseradish peroxidase (Bio-Rad) was used as a secondary antibody (1:2000). Blots were visualized by an enhanced chemiluminescence system (GE-Healthcare Bio-Sciences, Milan, Italy).

### 4.4. Cell Culture and Challenging Experiments

Caco-2 cells (ATCC HTB-37) were grown in a Dulbecco’s modified Eagle’s medium (DMEM) supplemented with 10% fetal bovine serum and grown at 37 °C in the presence of 5% CO_2_. Cells were seeded at a density of 1 × 10^5^ cell/mL in 35 mm cell culture dishes and grown for 48 h prior to being challenged with CE and EVs. Otherwise, cells were grown for 14 days (full-confluent and polarized monolayers) before being incubated with live L3 larvae. 

To ensure that the Caco-2 monolayer was not affected and disrupted by the larval mechanical action, we performed two-time points for ELISA incubation (6 h and 24 h) while for the CE and EVs challenge, a single time point at 24 h was assessed. To evaluate a potential temporal dynamic early host–pathogen interaction on the cytokines’ gene expression (Ebner et al., 2018) [51], three-time points at 1 h, 6 h, and 24 h for the qRT-PCR were assessed. After the indicated time points, cell supernatants were collected and stored at −80 °C for ELISA analyses, while cells were lysed and whole cell extracts were used for qRT-PCR analyses. 

### 4.5. Cytokines’ Measurements

Supernatants from cells challenged with live *Anisakis* L3, EVs, and CE were analyzed with ELISA assays (Thermo Fisher Scientific, Milan, Italy) to determine IL-6 and IL-8 amounts, according to the manufacturer’s protocol. 

### 4.6. Relative Quantification of Gene Expression by Real-Time PCR Analyses

Total RNA was obtained from Caco-2 cells using a TRIsure™ reagent (Bioline, London, UK), and contaminating genomic DNA was removed using a Turbo DNA-free kit (Life Technologies, Carlsbad, CA, USA). The amount of RNA was evaluated by spectrophotometric measurements using the Take3 module of the plate reader, BioTek SynergyHT and GEN5™, and the RNA integrity was evaluated through a run in an agarose gel 1.5%, stained with Syber safe (Invitrogen Waltham, Waltham, MA, USA). A total amount of 1µg RNA was reverse-transcribed for each sample, using SuperScript II RT (Invitrogen Waltham, Waltham, MA, USA) and OligodT (Invitrogen Waltham, Waltham, MA, USA) according to the manufacturer’s protocol. Target genes for the relative quantification by real-time PCR were IL-6 and IL-8, using GAPDH as the endogenous control. cDNA templates were mixed with a 2× PowerUp™ SYBR™ Green Master Mix (Applied Biosystem, Foster City, CA, USA) and specific primers, according to Borkowski et al. (Borkowski et al., 2014). Reactions included an initial holding stage of 2 min at 50 °C and of 2 min at 95 °C, followed by 40 cycles of PCR (95 °C, 15 s; 60 °C, 1 min); a final stage for melting curves was included to verify the specificity of the amplifications. A fold change in the expression level was calculated using the ∆∆Ct (Delta–Delta Ct) method. 

### 4.7. Statistical Analysis

ELISA protocols for IL-6 and IL-8 accounted for the calibrated reference, according to the manufacturer. The relative quantifications of Il6 and Il8 gene expression in RT-PCR were obtained using GADPH as a reference gene. The statistical significance of data was determined using a Student’s paired *t*-test to analyze statistical differences between two groups and a one-way analysis of variance (ANOVA) followed by a Bonferroni’s multiple comparison test to compare three or more groups. Statistical significance was considered from *p* ≤ 0.05.

## 5. Conclusions

In conclusion, the results here obtained showed an intricate interplay between the parasite and host; the live, active larva and its released EVs tried to silence the host’s immune response at the intestinal epithelium level to find a long-lasting niche to remain alive. Later on, a dying larva could induce the activation of the immune strategy of the host, leading to parasite expulsion, eosinophilia, and/or granuloma formation. Furthermore, the three challenges, representing different pathology phases, showed how the effects produced by *Anisakis* on the human host can be manifold, as also suggested in previous studies (Messina et al., 2016) [34]. In this scenario, it would be interesting to deeply investigate the molecular mechanisms at the base of this fine play of roles, expanding the research to other factors that could add important information about anisakiasis, as well as about the tumorigenic potential of these parasites.

## Figures and Tables

**Figure 1 pathogens-11-01214-f001:**
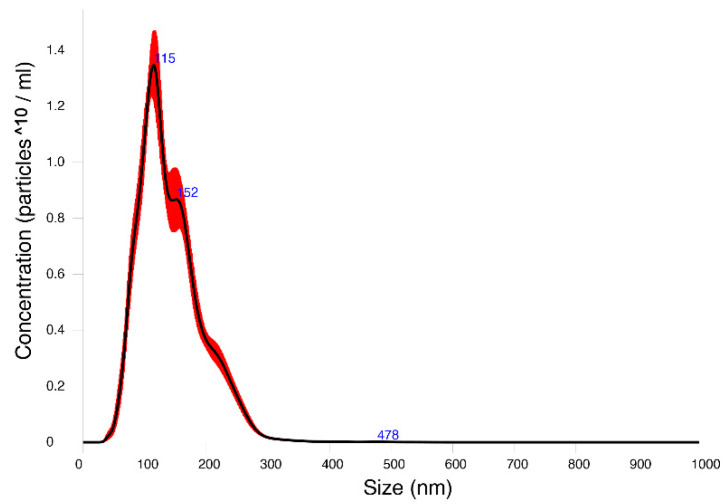
Finite track length adjustment (FTLA) concentration/size image for nanoparticle tracking analysis (NTA) of extracellular vesicles secreted by third-stage larvae of *Anisakis* spp. (*Anisakis simplex* sensu stricto, *Anisakis pegreffii*, and the hybrid form). The number of particles is intended as ^10.

**Figure 2 pathogens-11-01214-f002:**
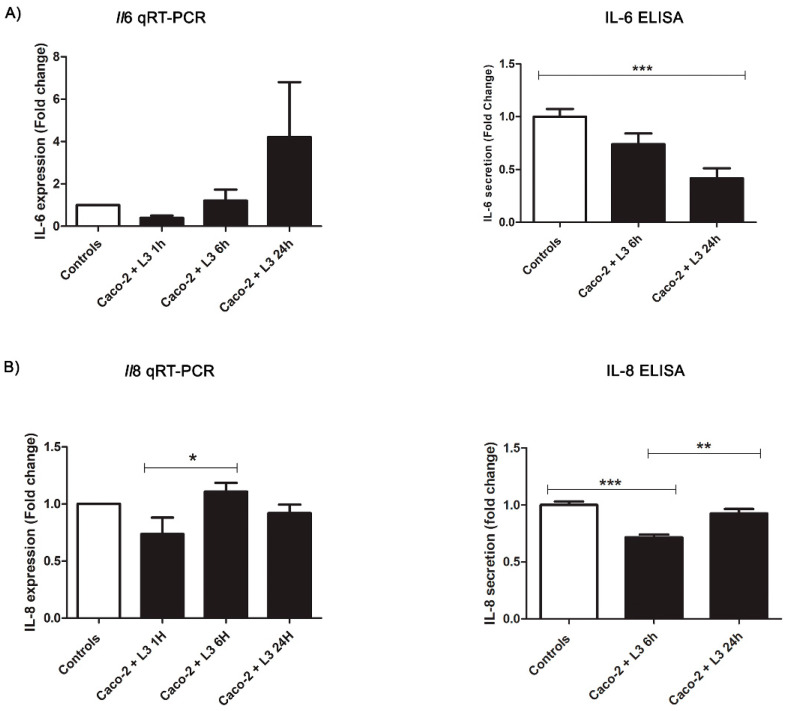
*Anisakis* live larvae (L3) modulation of cytokines’ secretion in Caco-2 cell monolayers after different times of incubation (1 h, 6 h, and 24 h). (**A**) *Il*6 gene expression and IL-6 levels in Caco-2 cell monolayers. (**B**) *Il*8 gene expression and IL-8 levels in Caco-2 cell monolayers. Data are expressed as a fold change compared to the control samples and as means ± SEM (standard error mean). Significance was evaluated using an ANOVA test and Bonferroni multiple comparison test. * *p* ≤ 0.05, ** *p* ≤ 0.01, and *** *p* ≤ 0.001.

**Figure 3 pathogens-11-01214-f003:**
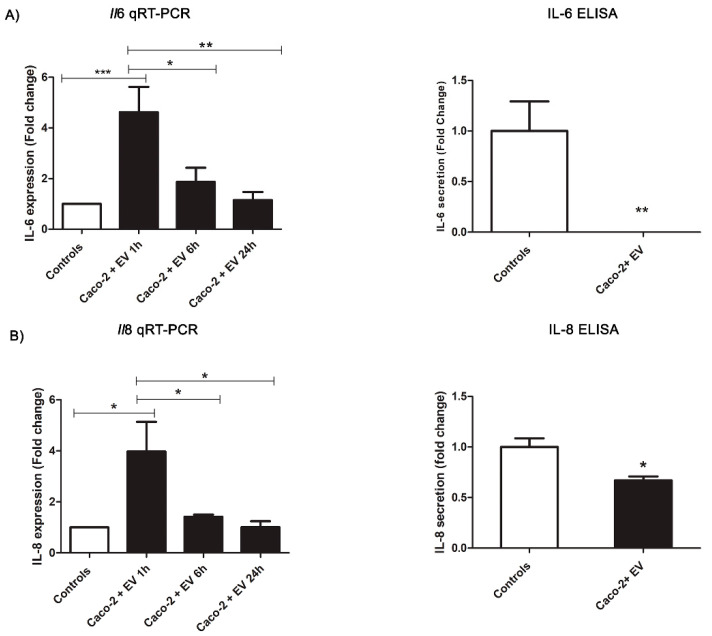
*Anisakis* extracellular vesicles (EVs) modulation of cytokines’ secretion in Caco-2 cell monolayers after different times of incubation (1 h, 6 h, and 24 h). (**A**) *Il*6 gene expression and IL-6 levels in Caco-2 cell monolayers. (**B**) *Il*8 gene expression and IL-8 levels in the Caco-2 cell monolayer. Data are expressed as a fold change compared to the control samples and as means ± SEM (standard error mean). Significance was evaluated using an ANOVA test and Bonferroni multiple comparison test and a Student’s *t*-test pairing for the Caco-2 cells’ controls vs. the Caco-2 cells exposed. * *p* ≤ 0.05, ** *p* ≤ 0.01, and *** *p* ≤ 0.001.

**Figure 4 pathogens-11-01214-f004:**
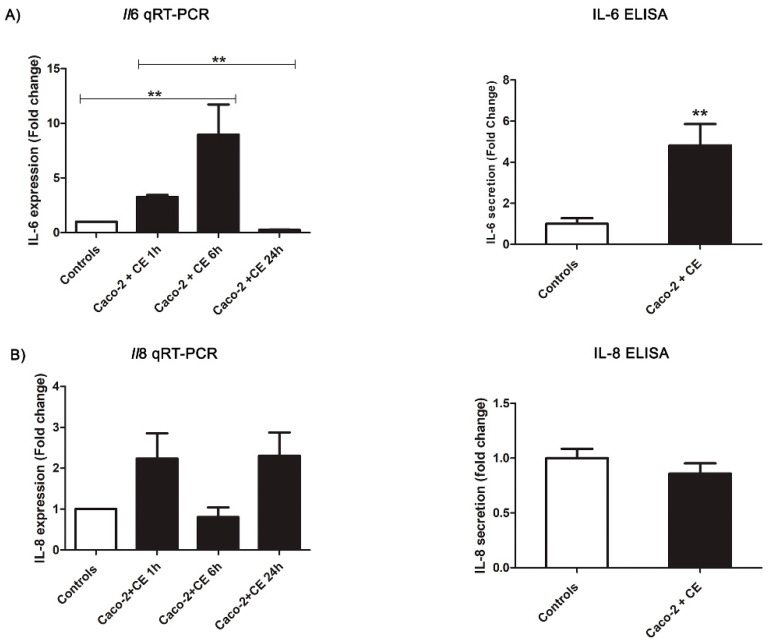
*Anisakis* crude extract (CE) modulation of cytokines’ secretion in Caco-2 cell monolayers after different times of incubation (1 h, 6 h, and 24 h). (**A**) *Il*6 gene expression and IL-6 levels in Caco-2 cell monolayers. (**B**) *Il*8 gene expression and IL-8 levels in Caco-2 cell monolayers. Data are expressed as a fold change compared to the control samples and as means ± SEM (standard error mean). Significance was evaluated using an ANOVA test and Bonferroni multiple comparison test and a Student’s *t*-test pairing for the Caco-2 cells’ controls vs. the Caco-2 cells exposed. ** *p* ≤ 0.01.

## Data Availability

Not applicable.

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
