# Peer review of "Inflammatory Response in Caco-2 Cells Stimulated with *Anisakis* Messengers of Pathogenicity"

_pathogens, 2022, doi:10.3390/pathogens11101214_

Round 1

Reviewer 1 Report

Abstract: please use past tense instead of present., e.g. The aim was…

Quantity/ concentration instead of amount

Induced/ triggered instead of determined

Gene and protein expression instead of protein production (change throughout the MS)

Please rephrase the sentence about cells not been alarmed, because readers do not know what you assume under alarmed cells.

Delete the in the activation

Response instead of strategy

That in vivo would lead to…. Instead of leading to

Introduction:

Since bivalves are also molluscs and yet are not infected with the genus, write cephalopods instead of molluscs.

Replace senescence with dying or similar term, as senescence is a physiological cell condition defined as biological aging. Larvae are not aging in this case, they are dying because of the host response. Please replace through the whole MS.

Line 47-48: Italian cases were diagnosed or in reality asymptomatic patients were hospitalised?

Line 53: you have listed no tapeworms, however only these 3 are listed as carcinogens group I and II, so avoid tapeworms.

Line 59: delete mucosal because it is not always related to mucosal layer

Line 61: scarce instead of poor

Line 83: given instead of thanks to

Line 92: quantity/ abundance/ concentration instead of amounts

Results:

Italics is missing 

Line 126: could the sequence be evolutionary longer in Anisakis as well? However, the difference between caco2 and CE/EVs is not negligible, and CE present another band at 100. Could you please explain this?

Line 139-140: if the observation is not stat. significant, do not characterise it (increased/decreased) as it is misleading.

Discussion:

I have no doubts that caco2 are useful model for parasites, but please tackle the fact that the cells are transformed and as such, physiologically very different from normal cells, and also in respose to pathogen. Acknowledging this fact should be enough, as everything has its pros and cons.

Line 207: instead of identified, tested or similar

Line 215: change determining 

Line 217: ESP do not ameliorate the immune response, rather they interact with it

Line 220: If you don’t define what alarmed cells are, this statement is very subjective. Also, you evaluated only 2 targets, so you don’t know whether on different level the cells were alarmed or not. Please rephrase.

Also, caco2 are not immune cells, so the whole physiological response to antigens is different and you cannot dismiss the possibility that this is their standard response. Line 221-224 is far fetching and if you wanted to test this hypothesis you should have used blood cells. Please change this paragraph. Line 236-238; Line 374-376: the same comment.

What I can overall see is that the proinflammatory response is balanced within 24h. Could authors shortly comment on this?

Methodology:

Line 298: rephrase, as it sounds like all collected larvae were immediately stored at -20 Cº.

Define what Alix is at the firs mentioning (full name, mono/polyclonal, rabbit, producer), and add the source of the atb in parenthesis.

Line 263-264: are there any miRNA in Anisakis EVs that might have affected the fate of Il6/8 gene expression in cells? Would that be too far fetching?

Line 322: undertaken instead of assessed.

Line 323: is there a number of caco2 cells used?

Line 328: whose antibody; mouse/goat/rat? Polyclonal I guess? Concentration?

Line 360: add the primer efficiency

Reviewer 2 Report

The manuscript entitled “Inflammatory response in Caco-2 cells stimulated with Ani-2 sakis messengers of pathogenicity” by Ilaria Bellini, Daniela Scribano, Meysam Sarshar, Cecilia Ambrosi, Antonella Pizzarelli, Anna Teresa Palamara, Stefano D’Amelio, and Serena Cavallero is devoted to investigation of human inflammatory response to different Anisakis vehicles of pathogenicity. It is an interesting paper dedicated to the study of interaction between host epithelial cells from humans and parasitic vehicles. This manuscript can be potentially published at the Pathogens journal, however, some critical issues should be addressed first.

Major points:

1. Why did the authors choose only pro-inflammatory cytokines? It is known that both types of immunity can play a role in parasitic diseases [10.3109/08830189809084491]. Is it possible that the strategy of the Anisakis is to induce anti-inflammatory response instead of pro-inflammatory?

2. Why the authors picked up only IL-6 and IL-8, but not tested TNFa or IL-1a cytokines, which are known to be ones of the most significant? Please, comment your choice.

3. The authors did not mention about the analysis of the integrity of RNA used in qPCR experiments. It is an extremely critical point of such experiments. Are the authors sure that the total RNA isolated from Caco-2 is not depredated? Please, provide the pictures of agarose gel electrophoresis with isolated RNA samples to the peers or in supplemental materials. If the authors did not perform this analysis, I would recommend the authors to add in the manuscript the information about this.

4. Did the authors perform the assessment of amplification rate using all primer pairs? ΔΔCt (Delta-Delta Ct) method is worked well only if the amplification rate of both, target and house-keeping gens, are comparable. Please, comment. So far, qPCR tests look unreliable.

 Minor points:

1. Figure 1 does not play a crucial role in the manuscript. It can be easily moved to the supplementary material.

2. Line 302: ”Pool of L3 (n=three) were used for obtaining crude extract samples. CE was prepared as follow: PBS 10X was added to each sample and then homogenated with pestles.”

Please provide the average weight of L3 per final volume of PBS (for instance, 0.1 or 0.01 mg/ml. It is a critical information for any extraction experiments.

3. The same in Line 309.

4. What does mean the first number part before p-value in brackets, like (F(2,37)= 8.143 P= 0.0001), (t(29)=3.42, P= 0.0065), etc.?

5. p-value is usually displayed as low-case italic p letter, not capital one. Please, correct through whole manuscript.

6. Line 275-277: “This pro-inflammatory enhancement could represent a trigger for Th2 response, able to activate and recruit immune cells (macrophages, DCs, eosinophils, neutrophils) to the site of damage”. Is it a mistake? Did the authors mean pro-inflammatory Th1-response instead of Th2?

7. I would recommend to place Conclusion section right after Discussion section.

Round 2

Reviewer 2 Report

The authors addressed all my questions.